# Application and Validation of LUXIE: A Newly Developed Virtual Reality Perimetry Software

**DOI:** 10.3390/jpm12101560

**Published:** 2022-09-22

**Authors:** Yen-Ting Chen, Po-Han Yeh, Yu-Chun Cheng, Wei-Wen Su, Yih-Shiou Hwang, Henry Shen-Lih Chen, Yung-Sung Lee, Su-Chin Shen

**Affiliations:** 1Department of Ophthalmology, Chang Gung Memorial Hospital, Taoyuan 333, Taiwan; 2College of Medicine, Chang Gung University, Taoyuan 333, Taiwan; 3Department of Ophthalmology, New Taipei Municipal Tucheng Hospital, New Taipei City 236, Taiwan

**Keywords:** head-mounted device, Humphrey, LUXIE, perimetry, virtual reality

## Abstract

Purpose: To report the application of LUXIE and validate its reliability by comparing the test results with those of Humphrey Field Analyzer 3 (HFA3). Methods: In this pilot study, we prospectively recruited participants who had received HFA3 SITA standard 30-2 perimetry and tested them with LUXIE on the same day. LUXIE is a software designed for visual field testing cooperating with HTC Vive Pro Eye, a head-mounted virtual reality device with an eye-tracking system. The test stimuli were synchronized with eye movements captured by the eye-tracking system to eliminate fixation loss. The global, hemifields, quadrants, glaucoma hemifield test (GHT) sectors, and point-by-point retinal sensitivities were compared between LUXIE and HFA3. All participants were asked to fill out a post-test user survey. Results: Thirty-eight participants with 65 eyes were enrolled. LUXIE demonstrated good correlations with HFA3 in global (*r* = 0.81), superior hemifield (*r* = 0.77), superonasal, superotemporal, and inferonasal quadrants (*r* = 0.80, 0.78, 0.80). The user survey showed that participants were more satisfied with LUXIE in operating difficulty, comfortability, time perception, concentration, and overall satisfaction. Conclusions: LUXIE demonstrated good correlations with HFA3. Fixation loss could be eliminated in LUXIE with the eye-tracking system. The application of virtual reality devices such as the HTC Vive Pro Eye makes telemedicine and even home-based self-screening visual field tests possible. Key Messages: 1. Virtual reality perimetry is a developing technology that has the potential in telemedicine, and home self-screening visual field tests. 2. LUXIE demonstrated good correlations with Humphrey Field Analyzer 3 in visual field retinal sensitivities.

## 1. Introduction

Standard automated perimetry (SAP) is an important method to evaluate the visual field of patients with ophthalmic or neurological diseases nowadays. Among contemporary SAPs, Humphrey (Carl Zeiss Meditec, Dublin, CA, USA) and Octopus (Haag-Streit AG, Koeniz, Switzerland) are the most commonly used perimeters as successors of Goldman perimetry [1,2]. Although SAPs have been well developed for decades, they still have several limitations [3]. First, it is difficult and uncomfortable for patients to keep their heads still in the perimeter bowl throughout the whole test, especially for those who are ill, claustrophobic, or physically limited. Second, SAP is prone to errors due to patients’ varying experiences and cooperativeness. During the entire examination, the patients’ fixation point may not stay still at all times. Although parameters such as fixation loss (FL), false positive (FP), and false negative (FN) help us distinguish the reliability of visual field tests, they cannot eliminate errors. Third, current SAP devices are bulky, heavy, space-occupying, and expensive, which largely limit the application of visual field in telemedicine, portable, and home-based healthcare.

To solve these problems, we developed a software named LUXIE, which is specially designed for visual field testing in collaboration with HTC Vive Pro Eye (HTC Corporation, Taoyuan, Taiwan), a virtual reality (VR) head-mounted device (HMD) with an eye-tracking system. It tracks the patient’s eye movement and adjusts the stimulus site instantaneously according to the current fixation point. The HMD is also portable and more comfortable for patients to wear during the examination. This study aims to validate the retinal sensitivity of LUXIE visual fields by comparing them with the retinal sensitivity of Humphrey Field Analyzer 3 (HFA3).

## 2. Materials and Methods

This prospective pilot study followed the Declaration of Helsinki and was approved by the Institutional Review Board of Chang Gung Memorial Hospital (CGMH), Taiwan. Written informed consent was obtained from all participants.

From July 2020 to January 2021, we recruited participants aged from 18 to 65 years old who had just received HFA3 examination (SITA standard 30-2, white on white, size III stimulus) at Linkou Chang Gung Memorial Hospital, Taiwan, and arranged LUXIE visual field examination on the same day. Participants with a history of epilepsy, facial deformities that could not wear HMD, or having cardiovascular electronic devices implantation were excluded. Participants who had unreliable HFA3 results (FL > 20%, FP > 15%, FN > 30%) were also excluded.

### 2.1. Hardware and Software Features

HTC Vive Pro Eye is a portable HMD that weighs 555 g and contains two active-matrix organic light-emitting diode screens in Fresnel lens (8.9 cm diagonal × 5.8 cm Width x 6.6 Height). It has a resolution of 1440 × 1600 pixels per eye, a refresh rate of 90 Hz, a maximal illumination of 125.73 cd/m^2^ (395 asb), and a field of view of 110° per eye. The interpupillary distance of the device is adjustable (60.7–73.5 mm). The built-in eye-tracking system, the Tobii eye tracker (Core SW 2.16.4.67) has an accuracy of 0.5°–1.1° and a sampling frequency of 120 Hz [4].

Like the Humphrey 30-2 program, the LUXIE test grids also contain 76 points in the central 30° visual field, separated 6 degrees apart, from 3 degrees to 27 degrees away from fixation (3°, 9°, 15°, 21°, 27°). Retinal sensitivity between 14 dB (display illumination 125.73 cd/m^2^, 395 asb) and 40 dB (display illumination 0.318 cd/m^2^, 1 asb) were tested at each point using a bisection strategy. That is, the test brightness of each test location is the median of the upper and lower limits of previous measurement. The test continues until the difference between the upper and the lower limit is less than 1 dB, which then LUXIE will take the median value as the final threshold sensitivity. The fixation point was tested first to establish a central threshold sensitivity which started from 27 dB (the median between 14 dB and 40 dB). The adjacent points were tested using the central threshold as the starting brightness in a random pattern. The Tobii eye tracker detected the current fixation point of the eye and displayed the stimuli exactly at the corresponding position. If the eye is moving during stimulus presentation, the stimulus will also move simultaneously according to eye movement. The stimulus presentation time was set as 200 milliseconds and the response time was set as 1 s. The background luminance was 10 cd/m^2^ (31.5 asb). The stimulus was 4 mm^2^, corresponding to the Goldmann size III stimulus (0.43°), and was displayed in a virtual bowl (radius 33 cm) using proper trigonometry adjustment during the examination.

The test result was presented as a RGB grayscale map. If the tested result was 14 dB, the map showed (0, 0, 0) black. If the tested result was 40 dB, the map showed (255, 255, 255) white. Results between 14 and 40 dB were shown linearly between RGB grayscale (1, 1, 1) and (254, 254, 254).

### 2.2. Experiment Procedures

The participants were first brought to a quiet room without outside interference where the computer, the HTC Vive Pro Eye HMD, and the base station were already set up. The HMD and controller were sanitized with tissues sprayed with 75% alcohol every time before applying to participants. The participants were then guided to comfortably sit on a chair facing the base station (Figure 1). The HMD was then put on the patient, and head straps were adjusted to better fit and center the patient’s field of vision. The embedded calibration system was used to calibrate pupil distance and eye-tracking performance for each participant before the examination (Figure 2a–c). If the calibration was not successful, the staff helped the participants adjust the wearing of HMD until the calibration passed. The participants could wear their glasses during the examination and no trial lens was used. Each eye was tested separately without eye patching. During the examination, the participants were asked to stare at the central fixation point, which was a green cross (Figure 2d), and to press the controller whenever they saw a visual stimulus. Position change or head movement was allowed. The participants were asked to fill out a user survey after completing the examination without the accompany of working staff. The operating difficulty, comfortability, time perception, concentration, and overall satisfaction were quantified using a Likert scale from 1 (worse) to 5 (best).

### 2.3. Statistical Analysis

The retinal sensitivities of LUXIE and HFA3 were averaged in global, each hemifield, each quadrant, and each glaucoma hemifield test (GHT) sector and analyzed with Pearson’s correlation coefficients (*r*). The retinal sensitivities of the 76 test locations in LUXIE and HFA3 were also analyzed with Pearson’s correlation coefficients individually. GHT sectors were defined as shown in Figure 3a [5]. Pearson’s correlation coefficients *r* ≥ 0.7 were defined as strong correlations; 0.4 ≤ *r* < 0.7 were defined as moderate correlations; *r* < 0.4 were defined as weak correlations [6]. Continuous variables were compared using the paired-sample *t*-test. All statistical analyses were performed using SPSS version 19.0 (IBM, Armonk, NY, USA) for Windows; *p* < 0.05 was considered statistically significant.

## 3. Results

Forty-three participants were recruited from July 2020 to January 2021. Among the 86 eyes, 21 were excluded because of poor HFA3 reliabilities (defined as FL > 20% or FP > 15% or FN > 30%). Overall, 65 eyes of 38 participants with 31 men and 7 women were enrolled in the analysis. The mean age of our participants was 50.2 ± 11.4 years. The mean test duration of LUXIE and HFA3 was 10.0 ± 1.4 and 7.3 ± 0.1 min per eye (*p* < 0.001).

The averaged HFA3 mean deviation (MD), pattern standard deviation (PSD), and visual field index (VFI) of our participants were −3.43 ± 4.2, 5.65 ± 4.5, and 90.4% ± 13%, separately. The global mean sensitivity of LUXIE showed strong correlation with HFA3 global mean sensitivity (*r* = 0.81, *p* < 0.001). The superior hemifield mean sensitivity of LUXIE also showed a strong correlation with that of HFA3 (*r* = 0.77, *p* < 0.001). The inferior hemifield mean sensitivity of LUXIE showed a moderate correlation with that of HFA3 (*r* = 0.65, *p* < 0.001). The superonasal, superotemporal, and inferonasal quadrant mean sensitivity of LUXIE showed strong correlations with that of HFA3 (*r* = 0.80, 0.78, 0.80; *p* < 0.001). The inferotemporal quadrant mean sensitivity of LUXIE showed moderate correlation with that of HFA3 (*r* = 0.41, *p* = 0.001). The correlations of mean retinal sensitivity of each GHT sector between LUXIE and HFA3 are shown in Figure 3b.

The correlation analysis of 76 test locations of LUXIE showed that 20 test locations were strongly correlated with that of HFA3, 32 test locations were moderately correlated, and 24 test locations were weakly correlated (Figure 4). Examples of HFA3 and LUXIE visual fields in the grayscale map are demonstrated in Figure 5. Eyelid artifact (Figure 5e) and rim artifacts (Figure 5f–h) were shown on HFA3 grayscale maps in some patients, but none on LUXIE.

The user survey showed that LUXIE achieved higher scores in all variables, including operating difficulty, comfortability, time perception, concentration, and overall satisfaction (Table 1).

## 4. Discussion

In this study, we successfully measured retinal sensitivity to light using LUXIE and HTC Vive Pro Eye VR HMD with the built-in Tobii eye tracker. Grayscale visual field maps were generated for each participant. LUXIE demonstrated strong correlation with HFA3 in global mean sensitivity (*r* = 0.81).

SAP is currently the gold standard for visual field testing. However, the equipment is heavy and bulky, thus it can only be performed in clinics or hospitals. The concept of using HMD to detect visual field defects was first proposed in 2000 to get rid of the cumbersome equipment of SAP. The portable head-mounted perimeter demonstrated equal sensitivity to confrontation visual field testing [7]. With the development of VR technology, the head-mounted eye-tracking perimeter VirtualEye was launched in 2014, which validated the concepts of head-mounted VR perimeter that used eye-tracking-based visual grasp instead of manual response [8]. Smartphone-based head-mounted perimeters were later introduced in 2017 and 2018 and showed promising results [3,9,10]. Perimeters based on VR technique were developed rapidly in recent years [11,12,13,14]. Another study using the Oculus Quest VR headset to simulate a standard visual field test was compared with the Octopus 900 and showed that the performance of the two was comparable [13]. In their study, correlation coefficients were 0.77, 0.50, and 0.70 for all, healthy, and glaucoma patients, respectively. VisuALL is another VR HMD with an eye-tracking system designed specifically for visual field testing [14]. Razeghinejad et al. showed that the global mean sensitivity of the VisuALL and the HFA had a good correlation in both normal (*r* = 0.5, *p* = 0.001) and glaucoma (*r* = 0.8, *p* < 0.001) groups. The mean sensitivity of all quadrants also had good correlations in both groups (*r* ranged from 0.3 to 0.8) [14]. Another HMD perimeter named imo^®^ (CREWT Medical Systems, Tokyo, Japan) can display a test target under the same test conditions as HFA. The mean sensitivity measured by imo^®^ showed high correlations with HFA (*r* ranged from 0.94 to 0.96) [15]. imo^®^ could generate advanced analysis such as MD, PSD, and VFI as in HFA. These parameters also showed high correlation with HFA (R^2^ > 0.81) [16].

In the current study, the correlation was moderate to weak in the inferior and temporal fields. As shown in Figure 4, the more inferior and temporal, the weaker the correlation. The reason for this trend is unclear, but possible explanations are as follows. First, the variability of the retinal sensitivity was higher in the peripheral visual field. It has been reported that the 30-2 Humphrey visual field, which contains the peripheral 24°–30° test locations, had higher variability than the 24-2 Humphrey visual field [17]. Other studies also showed that the variability increased rapidly as the observed sensitivity decreased. Peripheral test locations generally had lower sensitivities and therefore higher variability [18,19]. Second, our participants were allowed to wear their glasses during the test. The glasses frames might be pushed upward by the HMD during the test, which may affect the test results of lower or temporal VF (Figure 6) [20].

Although our results showed that the peripheral test locations had a weaker correlation with HFA3, it may not necessarily mean that the test result of LUXIE was inferior to that of HFA3. As shown in Figure 5, the lens rim artifact, which was not uncommon for HFA3, was not seen in LUXIE. This also explains in part the relatively poor agreement in the peripheral test locations. The test locations adjacent to the blind spot also showed weaker correlations in this study. It may be because that LUXIE did not have fixation loss of HFA3 since the built-in eye-tracking system in LUXIE adjusted the display of stimulus according to eye movement.

The user survey showed that the participants were more satisfied with LUXIE in terms of operating difficulty, comfortability, time perception, concentration, and overall satisfaction. Interestingly, although the actual test duration of LUXIE (10.0 ± 1.4 min) was significantly longer than that of HFA3 (7.3 ± 0.1 min), the participants thought that HFA3 took longer. The mandatory head and eye fixation during the Humphrey VF test usually caused severe discomfort and fatigue, which can explain this misperception of time.

In this study, we enrolled both healthy and glaucoma patients and performed a head-to-head comparison between LUXIE and HFA3. We believe that the device can not only benefit patients with glaucoma, but also patients with neurological (such as non-arteritic anterior ischemic optic neuropathy), retinal degenerative (such as retinitis pigmentosa), or intracranial diseases. Owing to its portability, LUXIE could also benefit bedridden patients and patients with head and neck restrictions who cannot fit in the traditional HFA3 machine. Another advantage of LUXIE is that it can be combined with the commercialized product HTC Vive Pro Eye without any additional equipment. In the future, people can just download this software to their own HTC Vive Pro Eye, which makes home-based visual field testing possible.

There were several limitations in this study. One major limitation of LUXIE is that the maximal light intensity of the headset screen was 125.73 cd/m^2^. Therefore, LUXIE is not able to detect retinal sensitivity below 14 dB. However, it can still be used to detect early-to-moderate glaucoma, hemianopsia from neurological causes, or for visual field screening. Second, the test duration was longer in LUXIE compared to HFA3 using the current bisection strategy. A better algorithm is needed in the future to shorten the test time without affecting its accuracy. Third, as a newly developed VR perimetry, LUXIE lacked the advanced analysis such as MD, PSD, and VFI provided by HFA3 that helps clinicians evaluate disease severity and progression. Fourth, as the participants in this study were voluntarily recruited, a possible bias might be that patients who were dissatisfied with HFA3 were more willing to try new VR HMD, causing the overwhelming results in the user survey. Lastly, the diagnostic agreement among LUXIE and HFA3 in glaucoma patients was not assessed. Future studies focusing on specific situations such as glaucoma, age, or level of education using a point-wise Bland-Altman approach are needed to explore the application of LUXIE in different clinical scenarios.

In summary, LUXIE/HTC Vive Pro Eye HMD is a portable, light, and comfortable device for VF testing. In this pilot study, LUXIE demonstrated good correlations with HFA3 in global, hemifields, quadrants, GHT sectors mean sensitivities, and point-by-point retinal sensitivity. The participants were more satisfied with LUXIE compared with HFA3. With the help of new technologies, the prospect of telemedicine and home-based visual field testing could be achieved with VR HMD in the future.

## Figures and Tables

**Figure 1 jpm-12-01560-f001:**
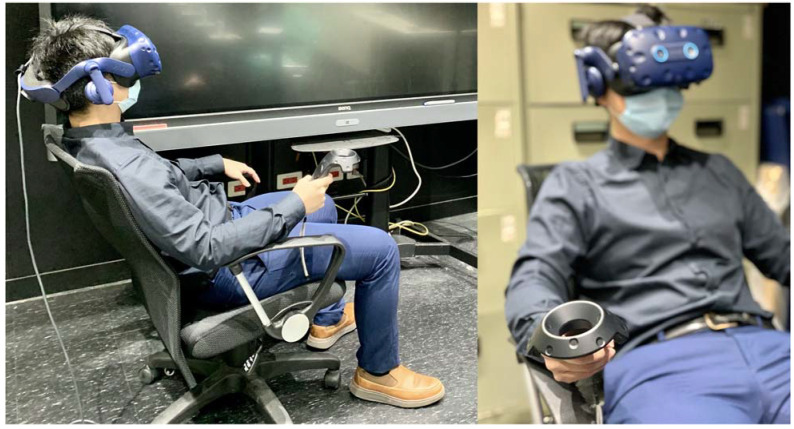
Demonstration of LUXIE/HTC Vive Pro Eye examination from the outside view.

**Figure 2 jpm-12-01560-f002:**
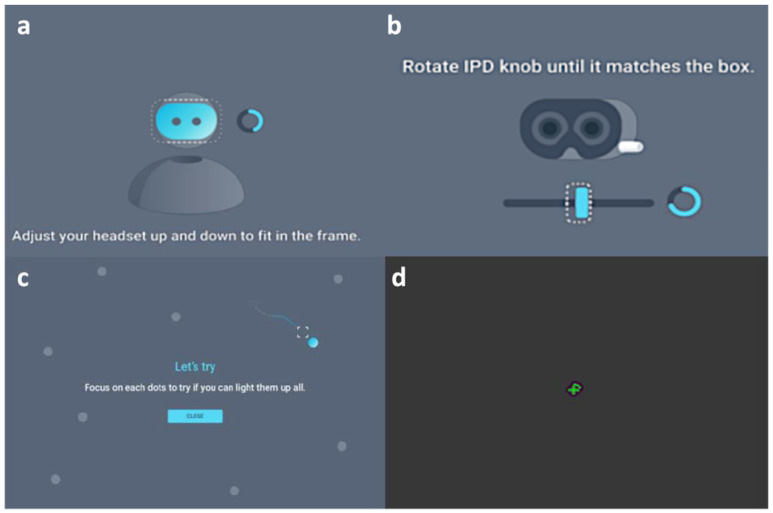
Demonstration of LUXIE/HTC Vive Pro Eye examination from the inside view. (**a**) Calibration of the headset position. (**b**) Calibration of the pupil distance. (**c**) Calibration of the eye-tracker (**d**), Demonstration of the central fixation green cross with a surrounding arc that will eventually become a circle along with the process of the test.

**Figure 3 jpm-12-01560-f003:**
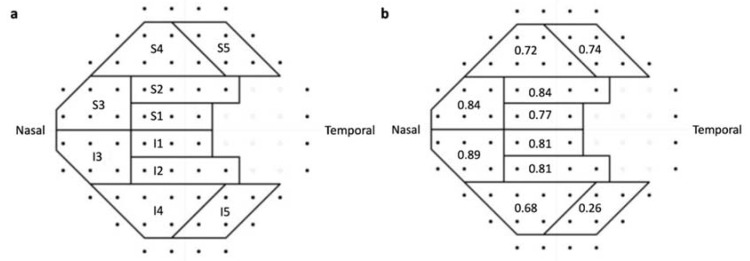
(**a**) The glaucoma hemifield test (GHT) sectors and its corresponding locations on central 30° visual field grids. (**b**) Correlations of GHT sectors between LUXIE and Humphrey Field Analyzer 3.

**Figure 4 jpm-12-01560-f004:**
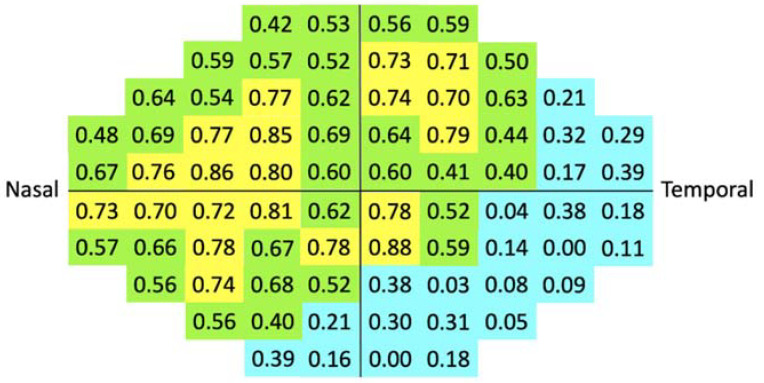
Correlations of 76 test locations between LUXIE and Humphrey Field Analyzer 3 shown on central 30° visual field grids. Strong (*r* ≥ 0.7), moderate (0.7 > *r* ≥ 0.4), and weak correlations (*r* < 0.4) were marked as yellow, green, and blue, separately.

**Figure 5 jpm-12-01560-f005:**
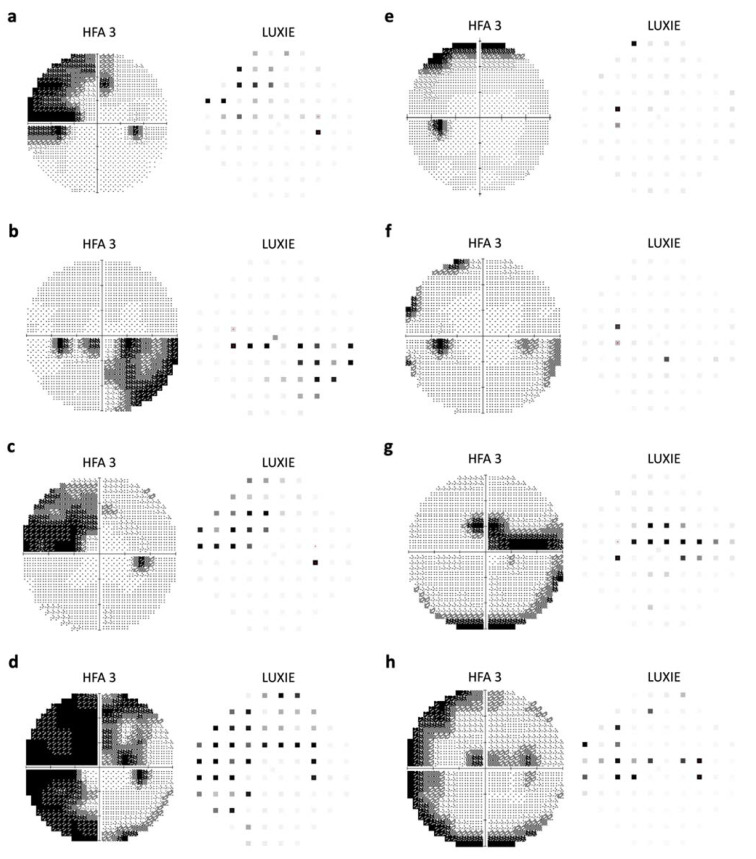
Examples of visual field grayscale maps generated by Humphrey Field Analyzer 3 (HFA3) and LUXIE. (**a**) Superonasal visual field defect. (**b**) Inferonasal visual field defect. (**c**) Superonasal visual field defect. (**d**) Superior arcuate scotoma and inferior nasal step. (**e**) Superior rim defect on HFA3. (**f**) Inferior nasal step; temporal rim defect on HFA3. (**g**) Superior arcuate scotoma; inferior rim defect on HFA3. (**h**) Superior arcuate scotoma; rim defect on HFA3.

**Figure 6 jpm-12-01560-f006:**
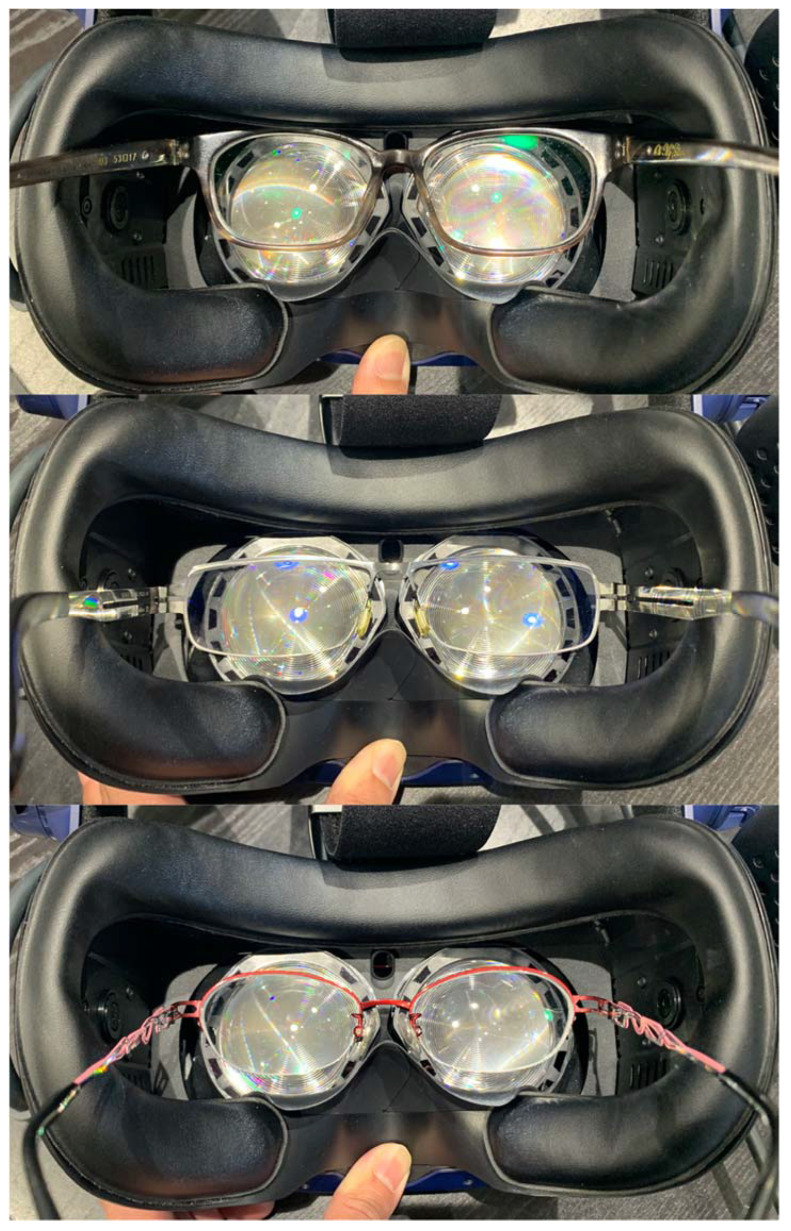
Simulation of the upward displacement of eyeglasses in different frames from an outside view.

**Table 1 jpm-12-01560-t001:** Analysis of the score between HFA3 and LUXIE in user survey (*n* = 38).

Score (1–5)	HFA3	LUXIE	*p* Value *
Operating difficulty	4.3 ± 1.0	4.8 ± 0.4	0.006
Comfortability	3.3 ± 1.0	4.4 ± 0.7	<0.001
Time perception	3.0 ± 0.9	3.6 ± 0.9	0.002
Concentration	3.7 ± 0.4	4.4 ± 0.4	<0.001
Overall satisfaction	3.5 ± 0.4	4.7 ± 0.4	<0.001

HFA3: Humphrey Field Analyzer 3. * *p* Value analyzed by Pair-sample *t*-test; *p* < 0.05 was considered to be statistically significant.

## Data Availability

The datasets generated or analyzed during the current study are available from the corresponding author on reasonable request.

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
