# Peer review of "Application and Validation of LUXIE: A Newly Developed Virtual Reality Perimetry Software"

_jpm, 2022, doi:10.3390/jpm12101560_

Round 1
Reviewer 1 Report
This virtual reality perimetry system is a very interesting idea.
Please emphasize it is a pilote study, and mention the weaknesses of the study.
The study is worth to continue: determine how the measurements depend on age, education and different pathologic situations
Author Response
Manuscript ID: jpm-1909700
Manuscript title: Application and Validation of LUXIE: A Newly Developed Virtual Reality Perimetry Software
Dear Editor-in-Chief,
We greatly appreciate the constructive comments of the reviewers and editors. We have now addressed these comments, which has strengthened the paper. We list our point-by-point responses below and have revised the manuscript accordingly. Each of the coauthors has seen and agrees with each of the changes made to this manuscript in the revision. After this revision, we hope you will deem this paper acceptable for publication in Journal of Personalized Medicine.
Reviewer #1:
This virtual reality perimetry system is a very interesting idea. Please emphasize it is a pilot study, and mention the weaknesses of the study. The study is worth to continue: determine how the measurements depend on age, education, and different pathologic situations
We appreciate this comment. We have made changes accordingly.
Action 1: In the Abstract, the description has been revised to “In this pilot study, we prospectively recruited participants who had received HFA3 SITA standard 30-2 perimetry and tested them with LUXIE on the same day.”
Action 2: In the Materials and Methods section, the description has been revised to “This prospective pilotstudy followed the Declaration of Helsinki and was approved by the Institutional Review Board of Chang Gung Memorial Hospital (CGMH), Taiwan.”
Action 3: In the Discussion section, the description has been revised to “There were several limitations in this study. One major limitation of LUXIE is that the maximal light intensity of the headset screen was 125.73 cd/m2. Therefore, LUXIE is not able to detect retinal sensitivity below 14 dB. However, it can still be used to detect early-to-moderate glaucoma, hemianopsia from neurological causes, or for visual field screening. Second, the test duration was longer in LUXIE compared to HFA3 using the current bisection strategy. A better algorithm is needed in the future to shorten the test time without affecting its accuracy. Third, as a newly developed VR perimetry, LUXIE lacked the advanced analysis such as MD, PSD, and VFI provided by HFA3 that helps clinicians evaluate disease severity and progression. Fourth, as the participants in this study were voluntarily recruited, a possible bias might be that patients who were dissatisfied with HFA3 were more willing to try new VR HMD, causing the overwhelming results in the user survey. Lastly, the diagnostic agreement among LUXIE and HFA3 in glaucoma patients was not assessed. Future studies focusing on specific situations such as glaucoma, age, or level of education using a point-wise Bland-Altman approach are needed to explore the application of LUXIE in different clinical scenarios.”
Action 4: In the Discussion section, the description has been revised to “In summary, LUXIE / HTC Vive Pro Eye HMD is a portable, light, and comfortable device for VF testing. In this pilot study, LUXIE demonstrated good correlations with HFA3 in global, hemifields, quadrants, GHT sectors mean sensitivities, and point-by-point retinal sensitivity.”
Reviewer #2:
I read the paper entitled “Application and validation of LUXIE: A new Developed Virtual Reality Perimetery Software” very carefully and concluded that the paper is acceptable with minor revision for publication in your journal. The topic of the article is interesting. The authors evaluated the possible role of virtual reality head-mounted device with an eye tracking system and compare the results of retinal sensitivity of LUXIE with the results of Humphrey Field Analyzer. Standard automated perimetry remains the gold standard in visual filed testing but portable head-mounted perimeter s could be another option, especially in environments where standard equipment was not available. Some minor corrections must be made by the authors. Regarding the literature, references no. 4,5, and 6 are not included in the text.
We appreciate this comment. We have assured references no. 4, 5, and 6 are included in the text and made some minor corrections. Reference no. 16 was also added and the following references were corrected accordingly.
Action 1: References no. 4, 5, and 6 were included in the text as follows.
- The built-in eye-tracking system, the Tobii eye tracker (Core SW 2.16.4.67) has an accuracy of 0.5°–1.1° and a sampling frequency of 120 Hz.[4]
- The retinal sensitivities of the 76 test locations in LUXIE and HFA3 were also analyzed with Pearson’s correlation coefficients (r) individually. GHT sectors were defined as shown in Figure 3a.[5]
- Pearson’s correlation coefficients r ≥ 0.7 were defined as strong correlations; 0.4 ≤ r < 0.7 were defined as moderate correlations; r < 0.4 were defined as weak correlations.[6]
Action 2: Reference no. 16 was added “Kimura T, Matsumoto C, Nomoto H. Comparison of head-mounted perimeter (imo®) and Humphrey Field Analyzer. Clin Ophthalmol. 2019 Mar 14;13:501-513. doi: 10.2147/OPTH.S190995.”
Sincerely,
Wei-Wen, Su, MD
Department of Ophthalmology,
Chang Gung Memorial Hospital, Taoyuan, Taiwan
5 Fu-Hsin Rd, Kweishan 333, Taoyuan, Taiwan.
TEL: 03-3281200 ext. 8666 ; FAX: 03-3287798
Email: vickysu@adm.cgmh.org.tw
Reviewer 2 Report
I read the paper entitled “Application and validation of LUXIE: A new Developed Virtual Reality Perimetery Software” very carefully and concluded that the paper is acceptable with minor revision for publication in your journal. The topic of the article is interesting. The authors evaluated the possible role of virtual reality head-mounted device with an eye tracking system and compare the results of retinal sensitivity of LUXIE with the results of Humphrey Field Analyzer. Standard automated perimetry remains the gold standard in visual filed testing but portable head-mounted perimeter s could be another option especially in environments where standard equipment was not available. Some minor corrections must be made by the authors. Regarding the literature, references no. 4,5, and 6 are not included in the text.
Author Response

(The authors gave the same response as above.)
